# Genomic Characterisation of the Relationship and Causal Links Between Vascular Calcification, Alzheimer’s Disease, and Cognitive Traits

**DOI:** 10.3390/biomedicines13030618

**Published:** 2025-03-03

**Authors:** Emmanuel O. Adewuyi, Simon M. Laws

**Affiliations:** 1School of Medical and Health Sciences, Edith Cowan University, Joondalup, WA 6027, Australia; 2Centre for Precision Health, School of Medical and Health Sciences, Edith Cowan University, Joondalup, WA 6027, Australia; s.laws@ecu.edu.au; 3Collaborative Genomics and Translation Group, School of Medical and Health Sciences, Edith Cowan University, Joondalup, WA 6027, Australia

**Keywords:** Alzheimer’s disease, abdominal aortic calcification, coronary artery calcification, cognitive traits, Mendelian randomisation, vascular calcification

## Abstract

**Background/Objectives:** Observational studies suggest a link between vascular calcification and dementia or cognitive decline, but the evidence is conflicting, and the underlying mechanisms are unclear. Here, we investigate the shared genetic and causal relationships of vascular calcification—coronary artery calcification (CAC) and abdominal aortic calcification (AAC)—with Alzheimer’s disease (AD), and five cognitive traits. **Methods:** We analyse large-scale genome-wide association studies (GWAS) summary statistics, using well-regarded methods, including linkage disequilibrium score regression (LDSC), Mendelian randomisation (MR), pairwise GWAS (GWAS-PW), and gene-based association analysis. **Results:** Our findings reveal a nominally significant positive genome-wide genetic correlation between CAC and AD, which becomes non-significant after excluding the *APOE* region. CAC and AAC demonstrate significant negative correlations with cognitive performance and educational attainment. MR found no causal association between CAC or AAC and AD or cognitive traits, except for a bidirectional borderline-significant association between AAC and fluid intelligence scores. Pairwise-GWAS analysis identifies no shared causal SNPs (posterior probability of association [PPA]3 < 0.5). However, we find pleiotropic loci (PPA4 > 0.9), particularly on chromosome 19, with gene association analyses revealing significant genes in shared regions, including *APOE*, *TOMM40*, *NECTIN2*, and *APOC1*. Moreover, we identify suggestively significant loci (PPA4 > 0.5) on chromosomes 1, 6, 7, 9 and 19, implicating pleiotropic genes, including *NAV1*, *IPO9*, *PHACTR1*, *UFL1*, *FHL5*, and *FOCAD*. **Conclusions:** Current findings reveal limited genetic correlation and no significant causal associations of CAC and AAC with AD or cognitive traits. However, significant pleiotropic loci, particularly at the *APOE* region, highlight the complex interplay between vascular calcification and neurodegenerative processes. Given *APOE*’s roles in lipid metabolism, neuroinflammation, and vascular integrity, its involvement may link vascular and neurodegenerative disorders, pointing to potential targets for further investigation.

## 1. Introduction

Alzheimer’s disease (AD) is a neurodegenerative disorder characterised by memory loss and progressive cognitive decline [1,2,3]. The growing burden of the disorder has become a major global public health concern, with over 10 million new dementia cases diagnosed annually, AD being the most prevalent form [1,2,3,4]. Although the exact causes of AD are not well understood, there is increasing appreciation of the likely role of vascular pathology in its onset and progression [5,6,7]. Emerging evidence highlights a link between vascular calcification and both AD and cognitive impairment [8,9,10]. Vascular calcification, including abdominal aortic calcification (AAC) and coronary artery calcification (CAC), refers to the abnormal deposition of calcium phosphate complexes within the walls of arteries, often serving as a marker of subclinical atherosclerosis [11].

Notably, AAC is widely prevalent, and evidence suggests that it strongly predicts several cardiovascular and respiratory conditions and independently increases the risk of myocardial infarction [12]. Similarly, CAC is commonly recognised as a critical indicator for predicting future atherosclerotic cardiovascular disease risk [13,14]. Atherosclerosis, the thickening and hardening of arterial walls due to cellular accumulation and calcification, contributes to the narrowing of blood vessels and is associated with cardiovascular diseases and neurodegenerative conditions, including dementia [6,15,16]. For instance, a study [16] recently highlighted the potential links between atherosclerosis and dementia, supporting the notion that systemic vascular health is closely tied to brain function. This observation reinforces the role of atherosclerosis as a potentially shared pathological factor in cardiovascular and neurodegenerative diseases.

More specifically, recent studies suggest an interaction between atherosclerosis-related traits and AD, potentially leading to additive effects on cognitive decline [7]. For example, an observational study reported that old women with AAC had an increased risk of developing all-cause late-life dementia [9]. The association between vascular calcification and cognitive impairment has also been observed in multiple populations [8,17,18,19]. Other studies, including cohorts from China and the Netherlands, observed associations between CAC and cognitive decline [8,10,17,18,19]. These findings suggest associations or co-occurring relationships between CAC or AAC and AD or cognitive impairment. Such relationships may accelerate disease progression and contribute to adverse outcomes, including more complex management plans [20]. On the other hand, the relationships may play a role in the pathogenesis of AD through shared genetic predisposition, common biological mechanisms, or direct causal relationships. Understanding the nature of these relationships and their underlying mechanisms could, thus, advance our knowledge of the aetiology of AD and provide opportunities for identifying therapeutic targets or precision preventative measures.

Despite growing evidence linking vascular calcification with AD and cognitive impairment, findings on this subject remain inconsistent. For instance, a systematic review and meta-analysis found limited evidence associating CAC scores with dementia risk [21], while a scoping review reported mixed results on the relationship between vascular calcification and AD or cognitive decline [22]. These inconsistencies, along with the limitations of traditional observational studies—such as measurement error, reverse causation, and residual confounding [23]—leave key questions unanswered. First, it remains unclear whether CAC or AAC is truly associated with AD or cognitive function, or if previous observational studies were biased or produced false-positive findings. Second, the nature of any observed relationships between AD and vascular calcification—whether causal or driven by shared genetic susceptibility—remains uncertain. Traditional observational studies are often limited in establishing causal relationships due to confounding influences from several factors, including lifestyles and environment [24,25,26,27]. In contrast, statistical genetic approaches (as in the current study) leveraging large-scale genome-wide association studies (GWAS) mitigate these limitations by using inherited genetic variants as instrumental variables. This genetic approach provides a more robust framework for investigating the genetic architecture underlying CAC, AAC, AD, and cognitive traits, with the potential to offer insights into their shared biological mechanisms [24,25,26,27]. Furthermore, genetic analyses can identify novel therapeutic targets by pinpointing variants and pathways contributing to disease risk, potentially guiding precision medicine strategies [24,25,26,27].

Hence, the current study systematically assesses the potential shared genetic and causal relationship of CAC and AAC with AD, cognitive performance, common executive function (cEF), and other cognitive-related traits. We analysed several large-scale genetic data employing well-regarded statistical genetics methods, including linkage disequilibrium score regression (LDSC), to examine genome-wide cross-traits genetic correlations. We also used the Mendelian randomisation (MR) method [28,29,30,31] to investigate whether CAC or AAC is causally associated with AD or cognitive traits, and vice versa. MR is a valuable and cost-effective method for assessing the relationship of CAC or AAC with AD or cognitive traits, as it uses genetic variants as instruments to infer causality while minimising the effects of confounding and reverse causation. Also, we utilised the pairwise GWAS (GWAS-PW) method to further investigate the relationship between CAC/AAC and AD/cognitive traits for insights into shared genetic variants and loci. Lastly, we performed gene-level analyses using a recently developed powerful gene-based association analysis method [32,33] to identify pleiotropic genes shared by CAC or AAC and AD or cognitive traits. The study approaches are especially important given the inconsistent evidence from traditional observational studies. By building evidence-based knowledge, and using a more robust genetic study, we can improve our understanding of the causes of AD and identify potential targets for further investigation.

## 2. Materials and Methods

### 2.1. Overview of Study Design

This study aimed to explore the genetic relationships of vascular calcification (CAC and AAC) with AD and cognitive traits. We employed four analytical approaches to achieve our objectives. First, we performed cross-traits genome-wide genetic correlation between CAC/AAC and AD/cognitive traits using the LDSC method. Given the strong evidence of the Apolipoprotein E (*APOE*) region’s significant impact on AD, we performed genetic correlation analyses excluding the region to determine if it drives any relationship between CAC or AAC and AD. Second, we assessed potential causal relationships through the bidirectional MR analysis method with sensitivity analyses and rigorous assessment of horizontal pleiotropy and heterogeneity. Third, we applied the GWAS-PW method to identify shared causal variants and pleiotropic loci for a pair of CAC or AAC and AD or cognitive traits, testing 1703 genomic regions.

The GWAS-PW method scans the genome to pinpoint pleiotropic loci by estimating the posterior probability that a genetic variant is causal for a pair of traits, or a genomic locus is associated with the two traits through distinct variants [34]. Lastly, we conducted gene-based association analyses to identify pleiotropic genes associated with CAC/AAC and AD/cognitive traits, prioritising pleiotropic loci identified in the GWAS-PW analysis. We used a recently developed and powerful gene-based analysis method, harnessing the strengths of three models of gene-based association analyses: fastBAT, mBAT, and mBAT-combo [32,33]. All analyses used well-powered GWAS summary data from individuals of European ancestry, with no evidence of significant sample overlap between the pairs of traits assessed.

During the preparation of this work, the authors used Grammarly and ChatGPT for grammar and language refinement. After using these tools, the authors reviewed and edited the content as needed and take full responsibility for the content of the publication.

### 2.2. Data Sources

We leveraged large-scale GWAS summary data from publicly available repositories and international research consortia. We sourced GWAS summary data for CAC from a recent study [35], which included 28,655 individuals. CAC was assessed using computed tomography imaging, quantified, and expressed using Agatston scores. This method is a standard non-invasive approach for evaluating the extent of calcified plaque in coronary arteries, which serves as a measure of subclinical atherosclerosis. For AAC, we used data from another recent publication, representing the largest-scale evaluation of AAC to date [12]. The study assessed AAC in 38,264 participants from the UK Biobank—a large population-based cohort of middle-aged to older adults (40–70 years) [12]. Lateral spine dual-energy X-ray absorptiometry scans, originally intended for bone mineral density assessment, were repurposed to evaluate AAC using manual scoring and machine learning methodologies [12]. Calcification in the lumbar spine (L1–L4) was graded using Kauppila’s scoring system [12,36]. Each vertebral segment was graded from 0 to 3: 0 for no calcification, 1 for calcification spanning less than one-third of the vertebra, 2 for one-third to two-thirds, and 3 for more than two-thirds. The scores for all four vertebrae were summed, yielding a total AAC score up to a maximum of 24 [12].

Additionally, we drew on one of the largest publicly available AD GWAS datasets, which comprised 71,880 cases and 383,378 controls, encompassing both clinically diagnosed AD and AD by proxy [37]. The clinically diagnosed AD dataset consisted of three case–control cohorts consisting of 79,145 individuals [37]. Participants were classified as cases or controls based on clinical diagnoses [37]. The ascertainment of AD cases was conducted through medical records or clinician-assessed diagnoses [37]. Control participants were screened to exclude neurodegenerative diseases, particularly dementia, ensuring they served as appropriate comparisons [37]. The ‘AD-by-proxy’ dataset was derived from the UK Biobank and included individuals whose biological parents had AD [37]. Information about the parents’ current age or age at death was included alongside the GWAS data. A high genetic correlation of 0.81 was observed between the ‘clinically diagnosed AD’ and ‘AD-by-proxy’ datasets, supporting their integration, as detailed in the original publication [37]. For potential (partial, due to data not being completely independent) replication, we used another AD GWAS data comprising only the clinically diagnosed cohorts [38]. AD was derived in the study through the meta-analyses of GWAS data from four consortia. The consortia included the Alzheimer’s Disease Genetic Consortium, Cohorts for Heart and Ageing Research in Genomic Epidemiology, European Alzheimer’s Disease Initiative, and the Genetic and Environmental Risk in Alzheimer’s Disease Consortium. The analysis included a large cohort of individuals of European ancestry, comprising 17,008 Alzheimer’s disease cases and 37,154 controls [38].

For cognitive traits, we included summary data from five studies: cEF (n = 427,037) [39], cognitive performance (n = 257,828) [40], intelligence (n = 269,867) [41], fluid intelligence scores (n = 125,935) [42], and educational attainment (n = 766,345) [40]. cEF refers to a unified cognitive factor that captures the shared variance across multiple executive function (EF) tasks, including response inhibition, interference control, working memory updating, and set shifting [39]. In the original publication, the authors conducted a GWAS analysis to derive a cEF factor score that represents the common executive control elements across various cognitive-related tasks [39]. Educational attainment in this study was measured as the number of years of schooling completed. This variable was uniformly defined across cohorts, with quality control applied to ensure consistency in the measurement. Cognitive performance was assessed using various tests designed to evaluate general cognitive abilities [40]. The specific tests and measures varied between cohorts but were standardised to enable meta-analysis across samples [40]. Intelligence was derived from various neurocognitive tests administered across multiple cohorts. These tests primarily measured fluid domains of cognitive functioning, such as reasoning, memory, and problem-solving [41].

In this study, we analysed quantitative variables from GWAS summary data as continuous traits, retaining their original scale as defined in the source studies. The summary statistics for each trait were derived from genome-wide association analyses, with effect sizes reported as beta coefficients. Detailed descriptions of the datasets, including participant demographics, study settings, measurements, and quality control procedures, are available in the referenced publications. Participants in all the data underlying our study were individuals of European ancestry. Appendix A offers a comprehensive overview of the data utilised in this study, including sample sizes and, where applicable, links to their sources.

### 2.3. Cross-Trait Genome-Wide Genetic Correlation Analyses

We conducted cross-trait genetic correlation analyses between CAC and each of AD and cognitive traits using the LDSC method. We similarly assessed the genetic correlation of AAC with AD and cognitive traits. The LDSC method estimates SNP heritability and genetic correlations by regressing GWAS test statistics (e.g., Z-scores) on LD scores for each SNP [43]. The method can distinguish true genetic signals from confounding factors and provide robust estimates of genetic relationships between traits [43].

Initially, we estimated genetic correlations with an unconstrained genetic covariance intercept to assess the potential proportion of sample overlap between traits, similarly to the practice in previous studies [26,27,43,44,45,46]. Our findings indicated that the genetic covariance intercepts were not significantly different from zero, suggesting no substantial sample overlap between AAC or CAC and AD or cognitive traits. Consequently, we proceeded with genetic covariance intercept-constrained analyses. In all LDSC analyses, pre-computed LD scores from the 1000 Genomes European reference panel were applied, excluding SNPs that did not intersect with the reference panel or had a MAF < 1%. To explore the potential impact of the *APOE* region (on AD-related analyses), we conducted analyses both with and without this region. A Bonferroni-corrected significance threshold of 8.33 × 10^−3^ was applied for genetic correlations involving AD and five cognitive traits, with *p*-values below 0.05 considered nominally significant.

### 2.4. Causal Relationship Assessment Using Mendelian Randomisation

We used the two-sample Mendelian randomisation (2SMR) analysis method to explore the potential causality of CAC or AAC with AD and various cognitive traits. Several MR methods were utilised, and we conducted a bidirectional assessment for a clear insight into the likely causal relationships of these traits. To ensure the robustness of our findings, we carefully selected appropriate instrumental variables (IVs) for the analysis and rigorously addressed potential issues around horizontal pleiotropy and heterogeneity [28,29,30]. We analysed data of the same ancestry (Europeans) and followed the STROBE-MR guidelines in this study’s design and conduct of MR analysis [31]. Figure 1 provides a summary of the procedural framework of our MR analysis.

#### 2.4.1. Selection of Instrumental Variables for MR Analysis

We selected IVs at the genome-wide significance (GWS) level (*p* < 5 × 10^−8^) from the relevant GWAS summary data used in our study. This stringent selection criterion ensures that the IVs are strongly associated with the exposure variables, with an F-statistic greater than 10 [28], thereby minimising the risk of weak instrument bias and fulfilling the first assumption of MR. Due to the limited availability of GWS IVs for CAC and AAC as exposure variables, we relaxed the selection threshold to the genome-wide suggestive level (*p* < 1 × 10^−5^). This adjustment was made with the understanding that using a smaller number of SNPs (<10) as IVs may introduce potential bias into our study. Although the second MR assumption—that IVs are not associated with confounders—is challenging or nearly impossible to fully validate, we rigorously evaluated our IVs and performed linkage disequilibrium clumping at a stringent threshold (*r*^2^ < 0.001, Figure 1) to enhance the independence of the selected instruments. Importantly, we ensured that our IVs were not linked to the outcome variables (*p* < 0.05), thereby contributing to adhering to the third MR assumption. To enhance the quality and reliability of our MR analysis, we excluded genetic variants with intermediate allele frequencies, as these can lead to ambiguous strand alignment. Additionally, we removed variants not present in the reference dataset and harmonised the exposure and outcome data, aligning alleles and ensuring consistency in effect directions. As highlighted in the sub-sections for the main MR and sensitivity analyses, we carried out other specific tests to ensure our IVs are robust.

#### 2.4.2. Performing MR Analyses

We used the inverse variance weighted (IVW) as the main MR model in the current study. This approach combines the Wald ratios for each genetic instrument into a single, weighted average, where the weights are the inverse of the variance of each ratio. The IVW method assumes the absence of horizontal pleiotropy, and the model is reliable so long as this assumption holds. To address potential heterogeneity among the causal estimates derived from different variants, we employed multiplicative random effects of the IVW model. Also, to complement our IVW estimates, we used additional MR methods, including the weighted median (which can yield valid estimates with up to 50% invalid IVs) and the MR-Egger (which can provide valid estimates by correcting for pleiotropy) [28,29]. We considered MR results with *p* < 0.05 as nominally significant and implemented a Bonferroni correction to mitigate the risk of false positives due to multiple tests across the various outcomes. Based on this adjustment, we set the significance threshold at *p* < 0.008 (0.05/6), corresponding to the analysis of six outcome variables (where applicable). We utilised the R statistical packages and the Unix environment for data management and analyses, and used the 2SMR software (version 0.5.6) [48,49], and MR-PRESSO [50] for the MR analyses (implemented on the R packages [version 4.2.1]).

#### 2.4.3. MR Sensitivity Analyses

Following the practice in related studies [28,51,52,53,54,55], and as part of the requirements for valid MR analysis [28,29,30,31], we conducted further tests to assess the reliability of our results. These tests include Cochran’s Q statistics to evaluate the heterogeneity of SNP effects, individual MR analyses, ’leave-one-out’ analyses to determine the impact of each IV on the overall results and examining the funnel plot for symmetry. We used the MR-Egger intercept to check for potential violations of the assumption of no unbalanced pleiotropy. Significant deviations of the MR-Egger intercept from zero indicate a possible violation of this assumption. Additionally, we implemented the MR pleiotropy residual sum and outlier (MR-PRESSO) method, which is known for identifying and addressing pleiotropy by excluding outliers [50]. Importantly, as an additional step, we meticulously reviewed our analyses and assessed each IV for their effect on the outcome variables. We excluded likely pleiotropic IVs associated with the outcome variables at a significance level of *p* < 0.05. This process involved scrutinising our LD clumped (clumping performed at *r*^2^  <  0.001) and properly harmonised data, excluding IVs with *P*_outcome-variable_ < 0.05, followed by MR analysis on the remaining instruments.

### 2.5. Assessing Shared Genetic Risk Loci: The Pairwise Gwas and Gene-Based Approach

We conducted colocalisation analysis using the GWAS-PW method, a tool designed to scan the genome for regions that likely share a causal variant or pleiotropic loci between traits [34]. This programme applies a Bayesian statistical model to estimate the probability (or PPA) and we modelled four potential scenarios: (1) a region contains a variant associated only with trait 1 (PPA1); (2) a variant associated only with trait 2 (PPA2); (3) a variant associated with both traits 1 and 2 (PPA3); or (4) independent variants are associated with each trait 1 and trait 2 (PPA4) but the region is shared by both traits [34]. In this study, we first conducted an analysis using CAC as trait 1 against AD and cognitive traits as trait 2. In the second analysis, we used AAC as trait 1 and, again, compared it with AD and cognitive traits as trait 2.

We applied GWAS-PW to assess potential shared causal variants and loci between CAC and AAC with AD and cognitive traits. The summary statistics for CAC and AAC were aligned with AD and cognitive trait data by rsID and alleles, ensuring consistent effect and non-effect alleles across traits. Standardised Z-scores and variances for each SNP were then used as the input to the GWAS-PW model [34]. The analysis covered 1703 predefined independent genetic regions based on LD patterns from the 1000 Genomes Project European reference data. There is no evidence for a significant overlap of samples between CAC or AAC and AD or cognitive traits, ruling out potential confounding or the need for adjustments in our analysis.

Our focus was on PPA3 and PPA4 results, thus, we considered regions with PPA3 > 0.9 to have a significant shared causal variant between the two traits (e.g., CAC and AD), while those with PPA3 > 0.5 were deemed suggestive. Similarly, we interpreted loci with PPA4 > 0.9 as harbouring distinct causal variants for each trait, influencing both independently, with the two traits sharing the loci. On the other hand, PPA4 > 0.5 was considered a suggestive association. The identification of risk variants or loci by GWAS-PW was further refined using three gene-based association analysis methods, including fastBAT, mBAT, and the mBAT-combo [32,33], to aggregate variant signals within genes and assess their overall contribution. Briefly, we aimed to detect shared genes in regions with strong evidence for pleiotropy based on PPA3/PPA4 > 0.9 (and at the suggestive level, PPA3/PPA4 > 0.5), and we used the gene-based method to ensure that the mapped genes were associated with the pairs of traits assessed. The mBAT-combo method combines mBAT and fastBAT statistics, and is superior to traditional sum-χ^2^ approaches, especially for identifying genes with masking effects [33]. The method has proven more powerful in simulations and real-world data [33]; hence, we prioritise using the method in the current study. For this analysis, SNPs were mapped within 50 kb of gene boundaries.

## 3. Results

### 3.1. Results of Genome-Wide Genetic Correlation Analyses

Table 1 presents the results of our genome-wide genetic correlation analyses. First, we observed a nominally significant positive genetic correlation (rg = 0.10, *p* = 3.14 × 10^−2^) between CAC and AD GWAS (from Jansen et al. [37]), but when excluding the *APOE* region, the signal weakens and becomes non-significant (*p* = 5.33 × 10^−2^, Table 1). We found no evidence of a significant genome-wide correlation between CAC and another AD GWAS (Lambert et al. [38]), with or without the exclusion of the *APOE* region. Conversely, we observed a negative correlation between CAC and cognitive traits, including cognitive performance (rg = −0.11, *p* = 1.59 × 10^−6^), fluid intelligence scores (rg = −0.11, *p* = 1.87 × 10^−5^), intelligence (rg = −0.11, *p* = 5.14 × 10^−6^), and educational attainment (rg = −0.10, *p* = 1.08 × 10^−6^), all surpassing the Bonferroni threshold. The correlation assessment between CAC and common executive function was not statistically significant (Table 1). Lastly, the genetic correlation between AAC and CAC was high (rg = 0.95, *p* = 4.70 × 10^−3^), surpassing the Bonferroni-corrected significance threshold, indicating a strong shared genetic basis between the two vascular calcification traits.

Second, we found no evidence of a significant genetic correlation between AAC and AD across different GWAS datasets, including the AD GWAS [37] (rg = 0.08, *p* = 2.32 × 10^−1^) and the AD GWAS [38] (rg = 0.01, *p* = 9.28 × 10^−1^). Excluding the *APOE* regions did not change these outcomes (Table 1). However, similar to CAC, we found significant negative genetic correlations of AAC with cognitive performance (rg = −0.10, *p* = 5.49 × 10^−3^) and educational attainment (rg = −0.13, *p* = 6.39 × 10^−5^), surpassing nominal and Bonferroni significance thresholds, respectively. The correlations with fluid intelligence (rg = −0.07, *p* = 5.24 × 10^−2^), intelligence (rg = −0.06, *p* = 1.12 × 10^−1^), and executive function (rg = −0.04, *p* = 2.53 × 10^−1^) were not statistically significant.

### 3.2. Results of MR-Based Causal Association Assessment

We conducted a 2SMR analysis to gain insights into the potential causal relationships between CAC or AAC and AD and cognitive traits. MR assesses causal effects under three specific assumptions: (1) the genetic variants used as IVs must have a strong association with the exposure, (2) these SNPs should be independent of any confounding factors, and (3) they should affect the outcome solely through their influence on the exposure [28,29,30,31]. We employed multiple MR approaches and performed a bidirectional analysis to gain a comprehensive understanding of these associations. To enhance the reliability of our results, we carefully selected suitable IVs and addressed concerns related to horizontal pleiotropy and heterogeneity. Figure 1 presents a detailed overview of our MR study design and highlights its specific assumptions.

The main MR results from the IVW model indicated no significant causal effect of CAC on AD (OR: 1.00, 95% CI: 0.99–1.01, *p*: 0.56) or AAC on AD (OR: 1.00, 95% CI: 0.98–1.03, *p*: 0.78) [Figure 2 and Figure 3, respectively]. Similarly, CAC or AAC had no significant causal effect on cognitive traits (Figure 2 and Figure 3). The only exception was the causal association of AAC on fluid intelligence scores. This effect was borderline nominally significant in the IVW model and nominally significant in both the weighted median and MR-PRESSO analyses (Figure 3 and Appendix A). However, the results did not survive the correction for multiple testing (Figure 3).

In the reverse analyses, our findings showed no significant causal effect of AD on CAC (OR: 1.29, 95% CI: 0.67–2.46, *p*: 0.44) or AAC (OR: 1.00, 95% CI: 0.86–1.16, *p*: 0.99) as summarised in Figure 4 and Figure 5, respectively. Likewise, none of the cognitive traits had a significant causal effect on CAC or AAC, except the borderline nominally significance of fluid intelligence scores’ effect on AAC (Figure 5), which did not survive correction for multiple testing. Current findings were consistent across other MR models, such as the weighted median and MR-Egger (Appendix A).

The MR-PRESSO method also found no significant relationship between AD and cognitive traits with CAC or AAC (except between fluid intelligence scores and AAC). Indeed, there were no outputs in the corrected analysis, suggesting the absence of outliers IVs. Appendix A provide comprehensive results for our MR analysis. MR-Egger intercept and heterogeneity tests indicate these findings were not biased by pleiotropic instruments (Table 2, and more comprehensively in Appendix A). Detailed information on IVs utilised for analysis is presented in Appendix A.

#### Addressing Horizontal Pleiotropy in Our MR Analysis

Here, we present findings of how we further addressed horizontal pleiotropy in our study—demonstrating how seemingly non-significant MR pleiotropy tests can lead to erroneous claims of causal associations. These results emphasise the importance of addressing pleiotropy and heterogeneity for valid causal estimates. For example, our initial analysis showed a significant causal effect of AD on CAC (IVW model, OR: 3.06, 95% CI: 1.60–5.85, *p*: 7.51 × 10^−4^, Appendix A). This result was supported by a non-significant pleiotropy test (Egger intercept: −0.020, *p*: 0.122) and corroborated by other MR methods, including MR Egger, the weighted median, and the weighted mode (Appendix A). Additionally, the crude estimate from MR-PRESSO aligned with the IVW result, supporting a significant causal effect of AD on CAC. The corrected MR-PRESSO results indicated a nominally significant causal influence of AD on CAC (OR: 2.11, 95% CI: 1.20–3.73, *p*: 1.68 × 10^−2^). Using the same approach, we found a significant causal effect of AD on AAC (Appendix A).

However, these results do not reflect true causal relationships, as there was evidence of significant heterogeneity (Appendix A). Upon examining the effect of the selected instruments on the outcome variables, we found that heterogeneity arose from the association of IVs with the outcomes, violating the third assumption of MR, which requires influencing the outcome only through the exposure pathway. IVs associated with the outcome variable—even at a nominal level—can directly affect the outcome, violating the exclusion restriction assumption. When we addressed the heterogeneity in our study by excluding pleiotropic SNPs, the results, which were once significant, became non-significant.

Based on our experience with recent MR publications, we speculate that such false positive results occur frequently, leading to erroneous causal claims or conclusions. The example in our study, thus, underscores the importance of rigorously addressing heterogeneity in 2SMR studies to avoid false claims of causality or misleading conclusions.

### 3.3. Shared Genomic Loci Between CAC or AAC and AD or Cognitive Traits

To advance our understanding of the relationship between CAC and AAC with AD and cognitive traits, we applied the GWAS-PW method [34] towards identifying potential pleiotropic loci or variants (see Section 2). Our analysis found that none of the tested 1703 genomic regions had a PPA3(the model where a shared locus with the same causal variant influences both traits [34]) greater than 0.5, indicating no evidence of causal SNPs associated with both CAC/AAC and AD/cognitive traits. However, we identified pleiotropic loci shared by CAC or AAC and AD or cognitive traits (Table 3 and Table 4), particularly on chromosome 19, suggesting a shared locus but with separate causal variants influencing the pair of traits. The PPA4 estimates were greater than 0.9 in these regions, indicating strong evidence or a high likelihood of pleiotropy, with the regions associated with both traits through distinct SNPs [34].

For example, our findings revealed that the locus at chr19:44,744,370–46,102,289 (hg19) is pleiotropic for CAC and AD, with separate top SNPs (Table 3, PPA4 =1). Similarly, the locus at chr19:44,744,108–46,102,684 (hg19) showed strong evidence of association, with AAC and AD involving distinct SNPs (Table 3). Using the mBAT-combo method, we identified genes within these regions, many of which exhibited significant associations (Bonferroni-adjusted *P*_gene_ < 1.06 × 10^−3^) with CAC or AAC and AD (or at least a nominal level of significance) (Table 3, PPA4 = 1), and were also identified by at least an additional gene-based method, either fastBAT or mBAT. These genes include *BCAM*, *TOMM40*, *NECTIN2*, *APOE*, *APOC1*, *CBLC*, *APOC4*, *APOC2*, *APOC4-APOC2*, *EXOC3L2*, and *CLPTM1* (Table 3).

Table 4 highlights the shared loci between CAC/AAC and cognitive traits and the pleiotropic genes associated with each trait pair. Notably, CAC and cEF share a region on chromosome 19: 44,744,370–46,102,547 (hg19, PPA4 = 1), where the implicated gene (*PHLDB3*) is only nominally significant for both traits. This locus was identified exclusively by the mBAT model. Similarly to AD, regions in chromosome 19 were significantly associated with CAC or AAC and some cognitive traits, including educational attainment, cEF, and fluid intelligence scores, as presented in Table 4.

Lastly, we identified additional loci with only suggestive association in the GWAS-PW analysis implicating regions in chromosomes one, six, seven, and nineteen, and their corresponding significant genes for the pair of traits assessed. Appendix A provides information about these loci and the likely pleiotropic genes implicated.

To improve the readability of the GWAS-PW’s findings and the interpretation of implicated genes, we summarise the results. Table 5 presents key pleiotropic loci with functional annotations. This table provides a biological context for the identified associations, helping to simplify the study’s interpretability. The functional annotations were derived using publicly available bioinformatics resources to verify gene functions, biological pathways, and disease associations. Specifically, gene functions were obtained from GeneCards (www.genecards.org, accessed on 21 February 2025), highlighting roles in lipid metabolism (*APOE*, *APOC1-APOC4*), mitochondrial function (*TOMM40*), immune response and cell adhesion (*NECTIN2*), and protein degradation (*CBLC*). Additionally, the literature on AD and cardiovascular traits was consulted to contextualise the biological relevance of these genes in vascular calcification and neurodegeneration.

## 4. Discussion

Using well-established methods, we present findings from analyses assessing the potential shared genetic architecture and causal relationships of CAC, and AAC, with AD, and cognitive traits. Our genome-wide genetic correlation analysis revealed a nominally significant association between CAC and AD. However, this result was not replicated in another AD dataset. While differences in study power may partially explain the variability, the observed nominal significance disappeared after excluding the *APOE* region, suggesting that the correlation initially observed was primarily driven by pleiotropy at this locus. Furthermore, we found no significant genome-wide genetic correlation between AAC and AD. Conversely, we found a significant negative genetic correlation between CAC and several cognitive traits, surviving correction for multiple testing, including cognitive performance, educational attainment, and intelligence scores. AAC also demonstrated a significant negative correlation with cognitive performance and educational attainment.

The significant negative genetic correlations observed suggest that a higher genetic predisposition to CAC and AAC is associated with lower cognitive performance. This finding aligns with studies reporting associations between vascular calcification and cognitive decline [8,17,18,19]. However, given the nature of genetic correlation assessments, this inverse relationship could also indicate that a higher genetic predisposition to better cognitive performance is linked to a reduced risk of vascular calcification. It is important to note that genetic correlation does not imply causation, as the observed associations may arise due to pleiotropy, shared genetic susceptibility, or other confounding factors. Additionally, for traits such as educational attainment, non-genetic influences—including socioeconomic conditions, environmental exposures, and social factors—may further complicate the relationship. Thus, while our findings indicate an inverse genetic association between CAC/AAC and certain cognitive traits, alternative explanations involving interactions between genetics, cognition, environment, and life-course factors may also play a role.

We performed bidirectional MR analyses to examine the potential causal associations of CAC and AAC with AD and cognitive traits and gain further insights into the nature of the relationships. Our comprehensive and rigorous analyses indicate that CAC or AAC is not causally associated with AD or cognitive traits, regardless of the direction of analysis, whether CAC or AAC were exposure or outcome variables. The only exception was the causal effect of AAC on fluid intelligence scores and vice versa, which was borderline nominally significant in the IVW model. These results did not survive correction for multiple testing, making them less convincing. Our findings were consistent across several MR models, and tests for heterogeneity or pleiotropy did not indicate potential bias. Vascular calcification, specifically CAC, is a well-established marker for improved risk prediction of subsequent atherosclerotic cardiovascular disease [13,14]. Atherosclerosis (characterised by vascular calcification) is associated with the risk of AD [6]. Indeed, a recent study highlighted an additive interaction effect between atherosclerosis and AD on cognitive functions [7]. Hence, it is reasonable to hypothesise that vascular calcification might be related to AD or cognitive traits. Several conventional observational studies support this hypothesis, suggesting a positive association between vascular calcification and AD or cognitive decline [9,10,17,18,19,56], although mixed results have also been reported [22]. Our genetic-based assessment, however, did not confirm convincing significant causal associations. Current results provide new insights into the interplay between these phenotypes and improve our understanding beyond what is known through traditional observational evidence.

A recent observational study, for example, reported an association between AAC and an increased risk of all-cause late-life dementia among elderly women [9]. While this study highlighted a potential link [9], our genetic-based analysis does not support a causal relationship. Unlike genetic approaches, traditional observational studies are often susceptible to unmeasured confounders and biases related to lifestyle or environmental factors, which may explain the observed associations. Additionally, our null MR results could stem from several factors, including the limitations of genetic instruments, or the possibility that the relationship between vascular calcification and AD involves complex disease mechanisms, such as mediation by other cardiovascular risk factors. Moreover, it should be noted that our study specifically focuses on AD, whereas the observational study [9] did not distinguish between dementia subtypes and was restricted to late-life onset (after 80 years) in women only. Future research, thus, should further investigate specific dementia subtypes and consider additional methodological approaches to better understand the interplay between vascular calcification, dementia, and cognitive decline.

Importantly, our study emphasises the necessity of rigorously addressing pleiotropy and heterogeneity to enhance the reliability of MR findings. Despite the challenges in proving some of them, it is essential to uphold the core assumptions of MR to infer valid causal estimates [30]. Adhering to best practices is vital to prevent estimating spurious causal associations, including implementing various MR approaches and conducting sensitivity testing [28,29,30,57]. Notably, heterogeneity tests can reveal potential biases in MR studies, as illustrated in our results. For example, there was evidence for heterogeneity between AD (as exposure) and CAC (as outcome) based on Cochran’s Q *p* value (Appendix A) in our illustrated example. This evidence signals a potential violation of MR assumptions and ignoring this observation would have resulted in false positives in our study. This premise, thus, underscores the need for further assessment, such as checking that the IVs are valid. We often found cases where authors overlook such significant heterogeneity tests in recent MR studies, potentially leading to claims of causality that may not be true.

The limited genome-wide genetic correlation and non-causal associations results in our analyses do not preclude shared genetic predisposition through specific regions in the genome. Hence, we progressed our study to investigate the potential shared variants or loci between CAC/AAC and AD or cognitive traits using the GWAS-PW and gene-based association methods. Our findings revealed that none of the 1703 tested genomic regions exhibited a PPA3 greater than 0.5, indicating a lack of evidence for causal SNPs influencing CAC or AAC and AD or cognitive traits. However, we identified loci on chromosome 19 demonstrating a high likelihood of pleiotropy (PPA4 > 0.9), indicating distinct causal variants affecting CAC/AAC and AD/cognitive traits. These findings implicate pleiotropic genetic underpinnings, primarily driven by the identified loci, particularly regions in chromosome 19, partly highlighting consistency between our genetic correlation and GWAS-PW results.

Within the implicated loci on chromosome 19, we identified several pleiotropic genes, including *BCAM*, *TOMM40*, *NECTIN2*, *APOE*, *APOC1*, *CBLC*, *APOC4*, *APOC2*, *APOC4-APOC2*, *EXOC3L2*, and *CLPTM1.* These genes exhibited significant associations with both CAC or AAC and AD or cognitive traits, indicating their role in the genetic architecture of these conditions and the potential co-occurrence of CAC/AAC with AD and cognitive decline. Recent findings further support the pleiotropic nature of chromosome 19 loci in AD-related traits and vascular phenotypes. For example, a study [58] highlights the involvement of *APOE* and its neighbouring genes (*TOMM40*, *NECTIN2*, *BCAM*) in AD and cognitive decline, partly reinforcing our findings of shared genetic susceptibility between vascular calcification and AD-related traits. The study [58] also underscores the importance of this region in complex trait interactions, aligning with our observation of pleiotropic effects. Furthermore, our analysis revealed shared loci between CAC/AAC and cognitive traits, as detailed in Table 4, including a significant region on chr19: 44744370–46102547 associated with CAC and cEF via *PHLDB3*. Although *PHLDB3* was only nominally significant (the only significant gene for both traits) and identified solely by the mBAT model of the gene-based analysis, the region exhibited a PPA4 = 1, indicating a high probability of pleiotropy (via distinct variants) for these traits.

Notably, the findings implicating *APOE* and its neighbouring genes as pleiotropic for AD and vascular calcification are intriguing. As the strongest genetic risk factor for late-onset AD [59], *APOE* ε4 exacerbates amyloid-beta accumulation, tau pathology, neuroinflammation, and age-related cognitive decline, contributing to neurodegeneration [60,61]. In contrast, *APOE* ε2 is considered protective in AD, enhancing amyloid-beta clearance and reducing amyloid plaque burden [62]. However, *APOE* ε2 is not entirely benign—it is linked to hypertriglyceridemia and type III hyperlipoproteinemia, which may increase cardiovascular risk in susceptible individuals [62]. Beyond its role in AD, *APOE* is a key regulator of lipid metabolism, though its cardiovascular effects vary by isoform. While *APOE* ε4 is associated with higher cardiovascular risk, exacerbating atherosclerosis and arterial calcification, *APOE* ε2’s role is more complex [62]. Although it may improve lipid transport and lower cholesterol levels in some cases, *APOE* ε2’s association with dyslipidaemia suggests a potential risk factor for cardiovascular disease under specific conditions [62,63,64].

Moreover, Apoe-knockout models indicate that *APOE* deficiency promotes vascular calcification through dysregulated lipid metabolism, chronic inflammation, and oxidative stress [61,65]. Apoe−/− mice further exhibit endothelial dysfunction, oxidative stress, and accelerated atherosclerotic plaque progression, reinforcing its role in vascular calcification [65]. Given *APOE*’s multiple roles, its shared genetic architecture may contribute to vascular calcification, AD pathology, and other metabolic or cerebrovascular conditions through distinct yet interconnected mechanisms. Such mechanisms may include dysregulated lipid metabolism, neuroinflammation, amyloid processing, impaired vascular integrity, and oxidative stress, collectively influencing both vascular and neurodegenerative disorders [61]. Additionally, *APOE* ε4 carriers experience increased cerebrovascular dysfunction, including blood–brain barrier disruption and impaired vascular integrity, which may link vascular calcification to neurodegeneration [66]. Future studies should further investigate *APOE*-mediated mechanisms to clarify their role in the interplay between vascular and neurodegenerative disorders.

In summary, our study provides new insights into the complex genetic relationships of CAC, and AAC, with AD, and cognitive traits. While we observed some genetic correlations—particularly a negative association between CAC/AAC and cognitive traits—our findings highlight the absence of causal relationships in the MR analyses. Further assessment using the GWAS-PW method suggests that potential associations do not result from shared causal SNPs but reflect genetic susceptibilities driven by distinct variants within implicated loci. Identifying pleiotropic loci and significant genes associated with vascular calcification, AD, and cognitive traits further supports this position.

### Strengths and Limitations

A key strength of this study is its use of a genetic approach, which is less susceptible to confounding from environmental and lifestyle factors, thus enhancing the reliability of our findings beyond what is possible through the conventional observational study approach. To our knowledge, this is the first study to investigate the relationship between vascular calcification, AD, and cognitive traits using statistical genetic methods. However, some limitations should be considered when interpreting our results. Firstly, the data utilised are exclusively from the European population, limiting the generalisability of our findings to other ancestries. Secondly, due to a lack of sufficient genome-wide significant instruments for CAC and AAC, we relaxed the threshold for IVs selection in our MR analysis to the suggestive level. This observation may indicate less powerful GWAS results for CAC and AAC, which could affect the non-significant causal associations observed in our study. We note, however, that we have used the best available data for CAC and AAC (to the best of our knowledge), in our study. Future research, thus, needs to explore these relationships further as more robust CAC and AAC GWAS data become available. Thirdly, although sample overlap can bias analyses such as genetic correlation, MR, and GWAS-PW, our preliminary evaluation indicates no evidence of significant sample overlap between CAC/AAC and AD/cognitive traits, thus ruling out bias from this factor in our study. Finally, our findings of limited genetic overlap and no causal association of CAC and AAC with AD or cognitive traits do not eliminate the possibility of associations due to shared risk factors. Therefore, comprehensive prospective studies may be beneficial for further elucidating the nature of these associations. Moreover, we recommend an assessment of the relationships based on the various types of dementia.

## 5. Conclusions

In conclusion, our study sheds new light on the intricate genetic relationships between CAC/AAC and AD/various cognitive traits. We observed notable genetic correlations, particularly a significant negative association between CAC/AAC and cognitive performance, which underscores the importance of genetic factors in these conditions. However, our findings from MR analyses highlight the absence of causal relationships between CAC or AAC and AD or cognitive traits, suggesting that the observed associations are not indicative of direct causal effects. Further assessments reinforce the notion that the genetic correlations observed are not due to shared causal SNPs, but rather reflect shared genetic susceptibilities influenced by distinct genetic variants within the implicated loci. The identification of pleiotropic loci and significant genes, such as those associated with the *APOE* region and other genes on chromosomes 1, 6, 7, 9, and 19, underscores the shared genetic susceptibility between vascular calcification, AD, and cognitive traits. These findings provide insights into the complex genetic interplay between the traits, revealing pleiotropic loci and significant genes that serve as essential targets for further investigation. Future research should focus on elucidating the mechanisms by which shared genetic factors influence both vascular and cognitive health. To build on our findings, future studies can integrate multi-omics approaches to uncover functional pathways linking vascular calcification and neurodegenerative processes. Experimental validation using cellular and animal models can be utilised to confirm the genetic associations and clarify their biological relevance. Additionally, given that genetic effects may vary by ancestry and sex, future research should investigate population-specific differences to improve the generalisability of findings.

## Figures and Tables

**Figure 1 biomedicines-13-00618-f001:**
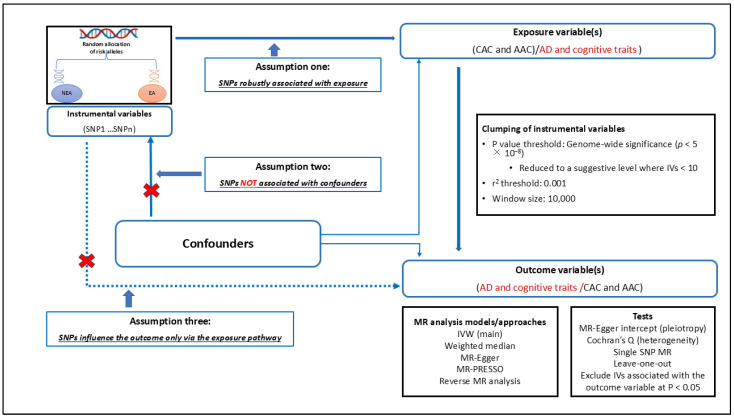
Schematic representation of MR assumptions and study design. The figure provides an overview of the MR analysis approach, emphasising its application and core assumptions in examining the potential causal relationship between exposure and outcome variables [28,29,30,31]. It highlights the three fundamental assumptions of MR: (1) the genetic variants (SNPs) used as instrumental variables must be robustly associated with the exposure, (2) these SNPs should not be associated with confounding factors (we note the direction of the arrow as recently illustrated [47]), and (3) they must influence the outcome exclusively through the exposure [28,29,30,31]. The figure also details the clumping parameters employed to ensure the independence and relevance of the genetic instruments. The analysis involves two rounds: first using CAC and AAC as exposures against AD and cognitive traits as outcomes, and the reverse analysis where AD and cognitive traits serve as exposures, with CAC and AAC as the outcome variables. AAC: abdominal aortic calcification, AD: Alzheimer’s disease, CAC: coronary artery calcification, EA: effect allele, IVW: inverse variance weighted, MR: Mendelian randomisation, MR-PRESSO: Mendelian randomization pleiotropy residual sum and outlier, NEA: non-effect allele, SNP: single nucleotide polymorphism.

**Figure 2 biomedicines-13-00618-f002:**
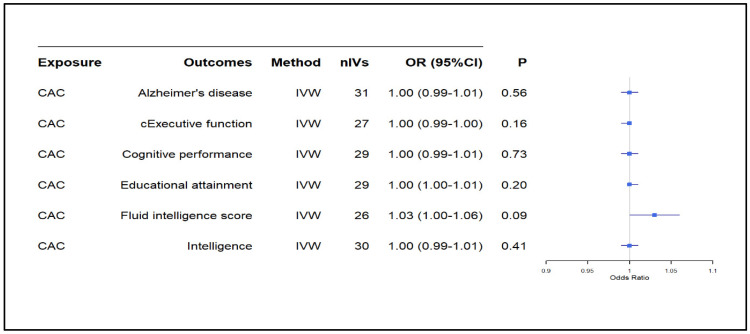
Causal effect of CAC on AD and cognitive traits. CAC: coronary artery calcification, CI: confidence interval, IVW: inverse variance weighted, nIV: number of instrumental variables, OR: odds ratio, *p*: *p*-value, cExecutive function: common executive function.

**Figure 3 biomedicines-13-00618-f003:**
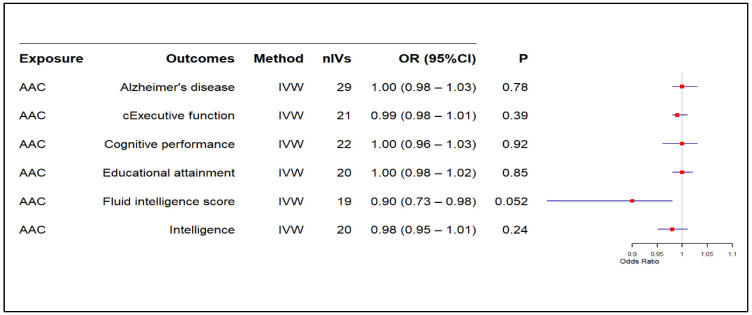
Causal effect of AAC onAD and cognitive traits. AAC: abdominal aortic calcification, CI: confidence interval, IVW: inverse variance weighted, nIV: number of instrumental variables, OR: odds ratio, *p*: *p*-value, cExecutive function: common executive function.

**Figure 4 biomedicines-13-00618-f004:**
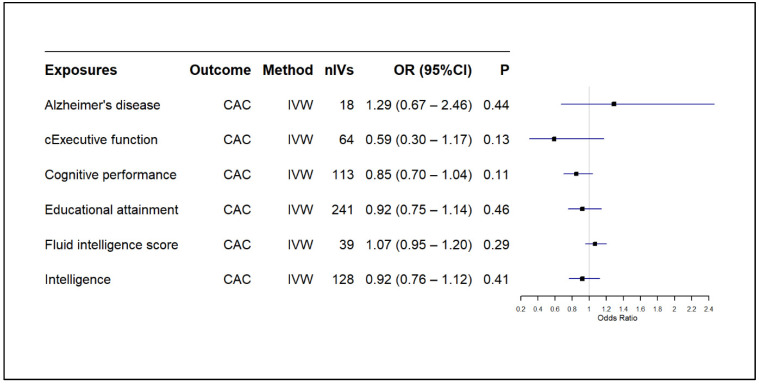
Causal effect of AD on cognitive traits against CAC. CAC: coronary artery calcification, CI: confidence interval, IVW: inverse variance weighted, nIV: number of instrumental variables, OR: odds ratio, *p*: *p*-value, cExecutive function: common executive function.

**Figure 5 biomedicines-13-00618-f005:**
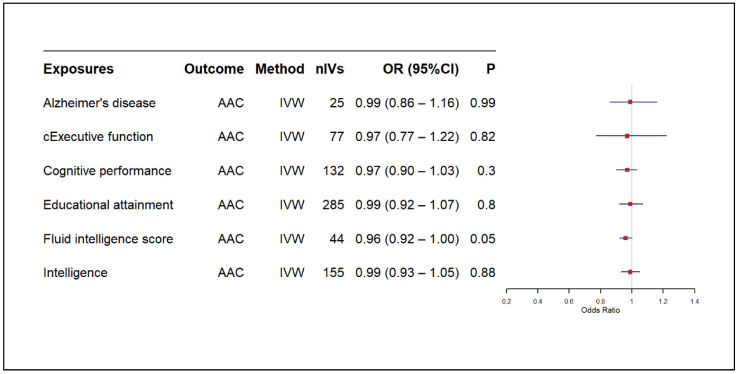
Causal effect of AD and cognitive traits on AAC. AAC: abdominal aortic calcification, CI: confidence interval, IVW: inverse variance weighted, nIV: number of instrumental variables, OR: odds ratio, *p*: *p*-value, cExecutive function: common executive function.

**Table 1 biomedicines-13-00618-t001:** Genome-wide genetic correlation between AAC, CAC, AD, and cognitive traits.

Trait 1	Trait 2	rg	se	*p*
CAC	AD GWAS [37]	0.1	0.05	3.14 × 10^−2^
	AD GWAS [37] excluding *APOE* region	0.1	0.05	5.33 × 10^−2^
	AD GWAS [38]	0.01	0.04	8.06 × 10^−1^
	AD GWAS [38] excluding *APOE* region	0	0.05	9.17 × 10^−1^
	Cognitive performance	−0.11	0.02	1.59 × 10^−6^
	Fluid intelligence scores	−0.11	0.03	1.87 × 10^−5^
	Intelligence	−0.11	0.02	5.14 × 10^−6^
	Common executive function	−0.04	0.02	7.55 × 10^−2^
	Educational attainment	−0.1	0.02	1.08 × 10^−6^
	Abdominal aortic calcification	0.95 *	0.34	4.70 × 10^−3^
AAC	AD GWAS [37]	0.08	0.07	2.32 × 10^−1^
	AD GWAS [37] excluding *APOE* region	0.07	0.07	3.49 × 10^−1^
	AD GWAS [38]	0.01	0.07	9.28 × 10^−1^
	AD GWAS [38] excluding *APOE* region	−0.01	0.07	9.19 × 10^−1^
	Cognitive performance	−0.1	0.03	5.49 × 10^−3^
	Fluid intelligence scores	−0.07	0.03	5.24 × 10^−2^
	Intelligence	−0.06	0.04	1.12 × 10^−1^
	Common executive function	−0.04	0.04	2.53 × 10^−1^
	Educational attainment	−0.13	0.03	6.39 × 10^−5^

AAC: abdominal aortic calcification, AD: Alzheimer’s disease, CAC: coronary artery calcification, GWAS: genome-wide association studies, APOE: Apolipoprotein E, rg: genetic correlation estimates, se: standard error, *p*: *p*-value. * estimated without constraining the genetic covariance intercept.

**Table 2 biomedicines-13-00618-t002:** MR sensitivity analyses of AD and cognitive traits with vascular calcification.

Exposure	Outcome	Heterogeneity Tests		Horizontal Pleiotropy Tests		
		Method	Cochran’s Q *p* Value	Method	Intercept	*p* Value
Coronary artery calcification vs. AD and cognitive traits						
CAC	Alzheimer’s disease	IVW	0.72	Egger intercept	−0.0013	0.72
	cExecutive function		0.94		−0.00042	0.59
	Cognitive performance		0.5		0.0013	0.43
	Educational attainment		0.81		0.00041	0.61
	Fluid intelligence scores		0.65		−0.0026	0.63
	Intelligence		0.39		0.00032	0.86
Abdominal aortic calcification vs. AD and cognitive traits						
AAC	Alzheimer’s disease	IVW	0.88	Egger intercept	−0.0014	0.55
	cExecutive function		0.71		−0.0019	0.11
	Cognitive performance		0.19		−0.0069	0.024
	Educational attainment		0.56		0.0013	0.47
	Fluid intelligence scores		0.69		−0.0057	0.5
	Intelligence		0.72		−0.0027	0.35
AD and cognitive traits vs. coronary artery calcification						
Alzheimer’s disease	CAC	IVW	0.26	Egger intercept	−0.021	0.11
cExecutive function			0.96		0.018	0.21
Cognitive performance			0.95		0.0097	0.89
Educational attainment			1		0.0022	0.7
Fluid intelligence scores			0.96		−0.019	0.45
Intelligence			0.86		−0.0014	0.88
AD and cognitive traits vs. abdominal aortic calcification						
Alzheimer’s disease	AAC	IVW	0.99	Egger intercept	−0.00019	0.95
cExecutive function			0.99		0.0047	0.55
Cognitive performance			0.95		0.0036	0.26
Educational attainment			1		−0.00096	0.63
Fluid intelligence scores			0.78		0.0065	0.88
Intelligence			0.98		−0.00017	0.95

AAC: abdominal aortic calcification, CAC: coronary artery calcification, IVW, inverse variance weighted, MR: Mendelian randomisation, *p*: *p*-value, cExecutive function: common executive function.

**Table 3 biomedicines-13-00618-t003:** Shared genomic loci of CAC and AAC with AD.

CAC/ AAC	AD	Chr: BP	PPA4	Shared Genes Gene	CAC/AAC			AD		
					Gene *P*_gene_ *	Top SNP	Top SNP *p*	Gene *P*_gene_ *	Top SNP	Top SNP *p*
CAC	AD	19: 44,744,370– 46,102,289	1.00	BCAM	7.08 × 10^−9^	rs118147862	1.40 × 10^−10^	0	rs41289512	1.5 × 10^−278^
				TOMM40	3.15 × 10^−8^	rs41290120	1.57 × 10^−11^	0	rs12972156	0
				NECTIN2	4.29 × 10^−8^	rs41290120	1.57 × 10^−11^	0	rs12972156	0
				APOE	2.69 × 10^−7^	rs41290120	1.57 × 10^−11^	0	rs12972156	0
				APOC1	8.13 × 10^−7^	rs41290120	1.57 × 10^−11^	0	rs12972156	0
				CBLC	1.01 × 10^−5^	rs118147862	1.40 × 10^−10^	3.67 × 10^−264^	rs41289512	1.5 × 10^−278^
				APOC4	1.99 × 10^−5^	rs7412	4.61 × 10^−10^	0	rs2075650	0
				APOC2	2.16 × 10^−5^	rs7412	4.61 × 10^−10^	0	rs10119	0
				APOC4-APOC2	2.97 × 10^−5^	rs7412	4.61 × 10^−10^	0	rs2075650	0
				EXOC3L2	4.56 × 10^−5^	rs12461144	7.03 × 10^−5^	1.95 × 10^−66^	rs10415850	2.17 × 10^−33^
				TRAPPC6A	5.08 × 10^−5^	rs12461144	7.03 × 10^−5^	4.30 × 10^−74^	rs28469095	1.07 × 10^−38^
				BLOC1S3	8.06 × 10^−5^	rs12461144	7.03 × 10^−5^	1.17 × 10^−65^	rs28469095	1.07 × 10^−38^
				NKPD1	3.92 × 10^−4^	rs10421247	1.04 × 10^−4^	1.06 × 10^−79^	rs28469095	1.07 × 10^−38^
				CLPTM1	6.14 × 10^−4^	rs7412	4.61 × 10^−10^	0	rs769449	0
				PPP1R37	1.59 × 10^−3^	rs10421247	1.04 × 10^−4^	5.47 × 10^−84^	rs28469095	1.07 × 10^−38^
				BCL3	3.23 × 10^−3^	rs148933445	1.26 × 10^−7^	7.93 × 10^−127^	rs2965169	9.24 × 10^−58^
				MARK4	5.66 × 10^−3^	rs12461144	7.03 × 10^−5^	3.44 × 10^−98^	rs28469095	1.07 × 10^−38^
				CEACAM16	8.13 × 10^−3^	rs62117204	1.58 × 10^−6^	7.75 × 10^−121^	rs2965169	9.24 × 10^−58^
AAC	AD	19: 44,744,108– 46,102,684	1.00	TOMM40	2.36 × 10^−11^	rs1065853	3.07 × 10^−13^	0	rs12972156	0
				NECTIN2	2.44 × 10^−10^	rs1065853	3.07 × 10^−13^	0	rs12972156	0
				APOE	2.55 × 10^−9^	rs1065853	3.07 × 10^−13^	0	rs12972156	0
				APOC1	3.97 × 10^−9^	rs1065853	3.07 × 10^−13^	0	rs12972156	0
				APOC2	6.12 × 10^−9^	rs1065853	3.07 × 10^−13^	0	rs10119	0
				APOC4	1.43 × 10^−8^	rs1065853	3.07 × 10^−13^	0	rs2075650	0
				APOC4-APOC2	1.55 × 10^−8^	rs1065853	3.07 × 10^−13^	0	rs2075650	0
				CLPTM1	2.55 × 10^−8^	rs1065853	3.07 × 10^−13^	0	rs769449	0
				BCAM	1.06 × 10^−5^	rs4803760	3.00 × 10^−7^	0	rs41289512	1.46 × 10^−278^
				CBLC	4.26 × 10^−4^	rs4803760	3.00 × 10^−7^	3.67 × 10^−264^	rs41289512	1.46 × 10^−278^

AAC: abdominal aortic calcification, CAC: coronary artery calcification, AD: Alzheimer’s disease, SNP: single nucleotide polymorphism, *p*: *p*-value. PPA4: posterior probability of association for model 4—the probability that a genetic locus is associated with both traits, signifying pleiotropy, but independent variants are associated with each trait. For instance, a PPA4 value > 0.90 indicates a high probability that the locus is pleiotropic for both traits. * Gene-based analysis was computed using the mBAT-combo approach, with results consistent with at least one additional gene-based method, either mBAT or fastBAT.

**Table 4 biomedicines-13-00618-t004:** Shared genomic loci of CAC and AAC with cognitive traits.

CAC/AAC	CT	Chr: BP	PPA4	Shared Genes	CAC/AAC			CT		
					Gene *p* *	Top SNP	Top SNP *p*	Gene-*p* *	Top SNP	Top SNP *p*
CAC	cEF	19: 44,744,370–46,102,547	1.00	** PHLDB3	4.38 × 10^−2^	rs62115754	2.28 × 10^−3^	3.88 × 10^−2^	rs11668385	1.10 × 10^−2^
CAC	EA	19: 44,744,370–46,102,547	0.99	TOMM40	3.15 × 10^−8^	rs41290120	1.57 × 10^−11^	7.56 × 10^−3^	rs405509	1.07 × 10^−5^
				NECTIN2	4.29 × 10^−8^	rs41290120	1.57 × 10^−11^	2.05 × 10^−2^	rs405509	1.07 × 10^−5^
				APOE	2.69 × 10^−7^	rs41290120	1.57 × 10^−11^	7.88 × 10^−3^	rs405509	1.07 × 10^−5^
				APOC1	8.13 × 10^−7^	rs41290120	1.57 × 10^−11^	6.22 × 10^−3^	rs405509	1.07 × 10^−5^
				APOC4	1.99 × 10^−5^	rs7412	4.61 × 10^−10^	5.83 × 10^−3^	rs405509	1.07 × 10^−5^
				APOC2	2.16 × 10^−5^	rs7412	4.61 × 10^−10^	1.01 × 10^−3^	rs405509	1.07 × 10^−5^
				APOC4-APOC2	2.97 × 10^−5^	rs7412	4.61 × 10^−10^	5.72 × 10^−3^	rs405509	1.07 × 10^−5^
				EXOC3L2	4.56 × 10^−5^	rs12461144	7.03 × 10^−5^	8.51 × 10^−5^	rs386569	8.22 × 10^−6^
				TRAPPC6A	5.08 × 10^−5^	rs12461144	7.03 × 10^−5^	3.77 × 10^−2^	rs12974200	3.56 × 10^−3^
				BLOC1S3	8.06 × 10^−5^	rs12461144	7.03 × 10^−5^	1.50 × 10^−3^	rs151165225	3.27 × 10^−5^
				CLPTM1	6.14 × 10^−4^	rs7412	4.61 × 10^−10^	4.07 × 10^−2^	rs405509	1.07 × 10^−5^
				PPP1R37	1.59 × 10^−3^	rs10421247	1.04 × 10^−4^	1.94 × 10^−2^	rs139290129	5.95 × 10^−4^
				MARK4	5.66 × 10^−3^	rs12461144	7.03 × 10^−5^	7.59 × 10^−7^	rs10402747	1.71 × 10^−8^
AAC	cEF	19: 44,744,147–46,101,600	1.00	TOMM40	2.36 × 10^−11^	rs1065853	3.07 × 10^−13^	1.57 × 10^−15^	rs429358	9.52 × 10^−20^
				NECTIN2	2.44 × 10^−10^	rs1065853	3.07 × 10^−13^	1.08 × 10^−14^	rs429358	9.52 × 10^−20^
				APOE	2.55 × 10^−9^	rs1065853	3.07 × 10^−13^	1.68 × 10^−16^	rs429358	9.52 × 10^−20^
				APOC1	3.97 × 10^−9^	rs1065853	3.07 × 10^−13^	1.18 × 10^−16^	rs429358	9.52 × 10^−20^
				APOC2	6.12 × 10^−9^	rs1065853	3.07 × 10^−13^	3.02 × 10^−16^	rs429358	9.52 × 10^−20^
				APOC4	1.43 × 10^−8^	rs1065853	3.07 × 10^−13^	1.02 × 10^−15^	rs429358	9.52 × 10^−20^
				APOC4-APOC2	1.55 × 10^−8^	rs1065853	3.07 × 10^−13^	7.48 × 10^−16^	rs429358	9.52 × 10^−20^
				CLPTM1	2.55 × 10^−8^	rs1065853	3.07 × 10^−13^	2.71 × 10^−15^	rs429358	9.52 × 10^−20^
				BCAM	1.06 × 10^−5^	rs4803760	3.00 × 10^−7^	2.63 × 10^−4^	rs4803764	4.24 × 10^−4^
				CBLC	4.26 × 10^−4^	rs4803760	3.00 × 10^−7^	1.78 × 10^−6^	rs12162222	6.16 × 10^−4^
AAC	EA	19: 44,744,147–46,101,600	0.96	TOMM40	2.36 × 10^−11^	rs1065853	3.07 × 10^−13^	7.56 × 10^−3^	rs405509	1.07 × 10^−5^
				NECTIN2	2.44 × 10^−10^	rs1065853	3.07 × 10^−13^	2.05 × 10^−2^	rs405509	1.07 × 10^−5^
				APOE	2.55 × 10^−9^	rs1065853	3.07 × 10^−13^	7.88 × 10^−3^	rs405509	1.07 × 10^−5^
				APOC1	3.97 × 10^−9^	rs1065853	3.07 × 10^−13^	6.22 × 10^−3^	rs405509	1.07 × 10^−5^
				APOC2	6.12 × 10^−9^	rs1065853	3.07 × 10^−13^	1.01 × 10^−2^	rs405509	1.07 × 10^−5^
				APOC4	1.43 × 10^−8^	rs1065853	3.07 × 10^−13^	5.83 × 10^−3^	rs405509	1.07 × 10^−5^
				APOC4-APOC2	1.55 × 10^−8^	rs1065853	3.07 × 10^−13^	5.72 × 10^−3^	rs405509	1.07 × 10^−5^
				CLPTM1	2.55 × 10^−8^	rs1065853	3.07 × 10^−13^	4.07 × 10^−2^	rs405509	1.07 × 10^−5^
AAC	FIS	19: 44,744,147–46,101,600	0.96	TOMM40	2.36 × 10^−11^	rs1065853	3.07 × 10^−13^	1.54 × 10^−3^	rs11668861	1.23 × 10^−3^
				NECTIN2	2.44 × 10^−10^	rs1065853	3.07 × 10^−13^	2.19 × 10^−3^	rs8113311	7.09 × 10^−4^
				APOE	2.55 × 10^−9^	rs1065853	3.07 × 10^−13^	3.20 × 10^−3^	rs11668861	1.23 × 10^−3^
				APOC1	3.97 × 10^−9^	rs1065853	3.07 × 10^−13^	6.93 × 10^−3^	rs11668861	1.23 × 10^−3^
				BCAM	1.06 × 10^−5^	rs4803760	3.00 × 10^−7^	3.39 × 10^−3^	rs8113311	7.09 × 10^−4^
				CBLC	4.26 × 10^−4^	rs4803760	3.00 × 10^−7^	2.41 × 10^−2^	rs8113311	7.09 × 10^−4^

AAC: abdominal aortic calcification, CAC: coronary artery calcification, cEF: common executive function, CT: cognitive traits, EA: educational attainment, FIS: fluid intelligence scores, SNP: single nucleotide polymorphism, *p*: *p*-value. PPA4: posterior probability of association for model 4—the probability that a genetic locus is associated with both traits, signifying pleiotropy, but independent variants are associated with each trait. For instance, a PPA4 value > 0.90 indicates a high probability that the locus is pleiotropic for both traits. * Gene-based analysis was computed using the mBAT-combo approach, with results consistent with at least one additional gene-based method, either mBAT or fastBAT. ** gene identified by mBAT gene-based model only.

**Table 5 biomedicines-13-00618-t005:** Key pleiotropic loci with functional annotations.

Locus (Chr:BP)	Key Genes	Associated Traits	Functional Annotation	Biological Implications
19: 44,744,370–46,102,289	*APOE*, *TOMM40*, *NECTIN2*, *APOC1*, *APOC2*, *APOC4*	CAC, AAC, AD, Cognitive Traits	*APOE* is a major lipid transport protein linked to AD risk; *TOMM40* is involved in mitochondrial protein transport; *NECTIN2* plays a role in cell adhesion and immune signalling	Associated with neurodegeneration, vascular health, and AD
19: 44,744,370–46,102,547	*PHLDB3*	CAC, Cognitive Traits (cEF)	Plays a role in cell signalling, potential involvement in neuronal function	May influence cognitive function and neurodevelopment
19: 44,744,108–46,102,684	*BCAM*, *CBLC*	AAC, AD	*BCAM* encodes a laminin-binding protein, implicated in cell adhesion; *CBLC* is involved in ubiquitin signalling and protein degradation	Suggests vascular contributions to AD risk through endothelial interactions
19: 44,744,147–46,101,600	*TOMM40*, *APOE*, *APOC1*, *NECTIN2*	AAC, Cognitive Traits (cEF, EA, FIS)	Overlaps with well-established AD risk loci, involved in lipid metabolism, mitochondrial function, and immune response	Supports shared genetic architecture between vascular calcification and cognition

AAC: abdominal aortic calcification, CAC: coronary artery calcification, cEF: common executive function, EA: educational attainment, FIS: fluid intelligence scores. Chr: chromosome, BP: base pair position.

## Data Availability

For this study, we utilised publicly available software for our analyses. Below, we provide the URLs for these tools, some of which include online methods and access to their respective details, such as documentation and (where applicable) source codes: GWAS-PW: https://github.com/joepickrell/gwas-pw (accessed on 19 September 2024), LDSC: https://github.com/bulik/ldsc (accessed on 24 March 2024), Two-Sample MR: https://mrcieu.github.io/TwoSampleMR/articles/introduction.html (accessed on 25 March 2024). The GWAS data used in this study were sourced from publicly accessible repositories and research groups or consortia, as specified in the data sources section. Appendix A contains further details about these datasets, including sample sizes and links to their sources. The published article and its Appendix A include all data generated during this study.

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
