# Peer review of "Genomic Characterisation of the Relationship and Causal Links Between Vascular Calcification, Alzheimer’s Disease, and Cognitive Traits"

_biomedicines, 2025, doi:10.3390/biomedicines13030618_

Round 1
Reviewer 1 Report
Comments and Suggestions for Authors
This manuscript investigates the genetic and causal relationships between vascular calcification (CAC and AAC), Alzheimer's disease (AD), and cognitive traits using genome-wide association studies (GWAS) and Mendelian randomization (MR). The findings contribute valuable insights into the genetic architecture and pleiotropic loci shared by vascular calcification and AD-related traits. However, several aspects require clarification or improvement for better clarity, impact, and reproducibility.
1. The reference cited on line 57 may not be comprehensive enough and lacks the latest relevant studies. It would be more appropriate to cite PMID: 38026222, as it aligns well with the content in this section.
2. Data and code need to be shared either through a code- sharing repo like GitHub or a docker- like system such as codeocean, so that other researchers can replicate the results.
3. The author is advised to polish the English.
4. The p letter for statistical analysis should be uppercase - italic face letter. Revise throughout the MS
5. Data and code need to be shared either through a code-sharing repo like GitHub or a docker-like system such as codeocean for clear reproducibility of the work.
6. line 602-612 lacks enough reference to valid these gene. It would be more appropriate to cite PMID: 37923804, as it aligns well with the content in this section.
Reviewer 2 Report
Comments and Suggestions for Authors
- This study employs multiple genetic approaches (LDSC, MR, GWAS-PW) with well-powered datasets, thorough sensitivity checks, and pleiotropy assessments, ensuring credible findings.
- The abstract should clearly state that no causal link was found to prevent misinterpretation.
- The introduction should better justify why a genetic approach is necessary over observational studies.
- The role of APOE in vascular calcification and AD should be discussed in more detail to provide context.
- The null MR results need further explanation, addressing possible reasons such as genetic instrument limitations or complex disease mechanisms.
- The readability of GWAS-PW findings could be improved by summarizing key pleiotropic loci in a concise table with functional annotations.
- Potential population differences, such as sex-specific or ancestry-related effects, should be acknowledged as a consideration for future research.
- The conclusion should be more impactful by suggesting concrete next steps, such as mechanistic studies or multi-omics integration, to enhance the study’s significance.

The language needs improvement.
Round 2
Reviewer 1 Report
Comments and Suggestions for Authors
Well revision but author should use STROBE-MR checklist to improve reporting of MR studies and cite PMID: 37198682 in line 231
Author Response
We sincerely appreciate the reviewer’s time and valuable feedback on our manuscript. Below are our responses to the comments provided:
- Use of STROBE-MR Checklist
Thank you for your suggestion. We have incorporated the STROBE-MR checklist to improve the reporting of our Mendelian randomisation (MR) study. The updated checklist aligns with the relevant sections of our manuscript to ensure comprehensive reporting. - Citation of PMID: 37198682
We have now cited PMID: 37198682 at line 231, as recommended. This citation provides additional context and supports the reporting of our MR study.
We appreciate the reviewer’s positive evaluation of the quality of the English language in our manuscript. Thank you once again for your constructive feedback, which has helped enhance the clarity and reporting of our study.